# Multiple guidance mechanisms control axon growth to generate precise T-shaped bifurcation during dorsal funiculus development in the spinal cord

Bridget M Curran[1], Kelsey R Nickerson[2,3], Andrea R Yung[4], Lisa V Goodrich[4], Alexander Jaworski[2,3], Marc Tessier-Lavigne[5], Le Ma[1]*

[1]Department of Neuroscience, Jefferson Synaptic Biology Center, Vickie and Jack Farber, Institute for Neuroscience, Sydney Kimmel Medical College, Thomas Jefferson University, Philadelphia, United States; [2]Department of Neuroscience, Brown University, Providence, United States; [3]Robert J. and Nancy D. Carney Institute for Brain Science, Providence, United States; [4]Department of Neurobiology, Harvard Medical School, Boston, United States; [5]Department of Biology, Stanford University, Stanford, United States

*For correspondence:
le.ma@jefferson.edu

Competing interest: The authors declare that no competing interests exist.

**Abstract** The dorsal funiculus in the spinal cord relays somatosensory information to the brain. It is made of T-shaped bifurcation of dorsal root ganglion (DRG) sensory axons. Our previous study has shown that Slit signaling is required for proper guidance during bifurcation, but loss of Slit does not affect all DRG axons. Here, we examined the role of the extracellular molecule Netrin-1 (Ntn1). Using wholemount staining with tissue clearing, we showed that mice lacking Ntn1 had axons escaping from the dorsal funiculus at the time of bifurcation. Genetic labeling confirmed that these misprojecting axons come from DRG neurons. Single axon analysis showed that loss of Ntn1 did not affect bifurcation but rather altered turning angles. To distinguish their guidance functions, we examined mice with triple deletion of Ntn1, Slit1, and Slit2 and found a completely disorganized dorsal funiculus. Comparing mice with different genotypes using immunolabeling and single axon tracing revealed additive guidance errors, demonstrating the independent roles of Ntn1 and Slit. Moreover, the same defects were observed in embryos lacking their cognate receptors. These in vivo studies thus demonstrate the presence of multi-factorial guidance mechanisms that ensure proper formation of a common branched axonal structure during spinal cord development.

## eLife assessment

This **important** study expands our understanding of the role of two axon guidance factors in a specific axon guidance decision. The strength of the study is the **compelling** axonal labeling and quantification, which allows the authors to establish precise consequences of the loss of each guidance factor or receptor.

## Introduction

With only one axon extending out from the soma, a neuron uses axonal branches to connect with multiple synaptic targets in complex neural circuits. Often axonal branches develop stereotypic patterns that support proper circuit connection and function. This involves not only the generation of new branches at the right location but also the guidance of newly formed branches toward their

targets (*Gibson and Ma, 2011*; *Kalil and Dent, 2014*). While extensive knowledge has been gained on the molecular mechanisms of axon guidance and axon branching in the past (*Armijo-Weingart and Gallo, 2017*; Chédotal, 2019; *Gallo, 2011*), less attention has been paid to how branches are guided in association with branch formation when precise circuits are assembled.

The central projections of the dorsal root ganglia (DRG) sensory neurons in the spinal cord provide an excellent model to investigate such guidance mechanisms during branch morphogenesis. These projections relay somatosensory information (e.g., pain and touch) collected by their peripheral projections in the skin and muscle to the brain (*Altman and Bayer, 1984*; *Mirnics and Koerber, 1995*; *Nascimento et al., 2018*; *Ramon y Cajal, 1904*). Upon reaching the dorsal spinal cord, they normally bifurcate in a location called the dorsal root entry zone (DREZ; *Mirnics and Koerber, 1995*; *Ozaki and Snider, 1997*). The bifurcated branches appear to extend in opposite directions along the rostrocaudal axis, generating T-shaped junctions, as part of the dorsal funiculus in the DREZ (*Figure 1E*; *Gibson and Ma, 2011*; *Nascimento et al., 2018*). The dorsal funiculus is a critical axonal track in the spinal cord that allows information to flow between the peripheral nervous system (PNS) and the central nervous system (CNS). Damage to or malformation of this evolutionarily conserved axonal structure in the DREZ can lead to paralysis, as commonly seen after spinal cord injury or in genetic disorders such as Frederick ataxia (*Koeppen et al., 2017*; *Smith et al., 2012*; *Zheng et al., 2019*). Thus, understanding the development of DRG axon bifurcation could provide useful insights into the mechanisms that ensure the fidelity of forming branched circuits.

Studies in the past have begun to tease out the molecular and cellular mechanisms that generate this stereotypic branch. In mice, after reaching the dorsal spinal cord at embryonic day (E) 10.5 (*Mirnics and Koerber, 1995*; *Ozaki and Snider, 1997*), the DRG central projection first generates a new branch in response to the C-type natriuretic peptide (CNP) present in the dorsal spinal cord, a step that requires the CNP receptor Npr2 as well as cGMP signaling (*Schmidt et al., 2009*; *Schmidt et al., 2007*; *Zhao and Ma, 2009*; *Zhao et al., 2009*). The resulting two daughter branches are then guided to grow only in the DREZ along the rostrocaudal axis (*Ozaki and Snider, 1997*), leading to the formation of the T-shaped branch junction. This step is partly controlled by the Slit family of guidance molecules, Slit1 and Slit2, which are expressed next to the DREZ inside the spinal cord, as well as their Robo receptors, which are expressed by DRG neurons (*Ma and Tessier-Lavigne, 2007*). In mouse mutants lacking Slit1 and Slit2 or their receptors Robo1 and Robo2, one of the daughter branches misprojects into the spinal cord. However, the defect only affects ~50% DRG axons and does not eliminate the dorsal funiculus, suggesting the presence of other molecular mechanisms that guide bifurcating DRG afferents.

In search of additional molecular mechanisms, we re-evaluated the role of Netrin-1 (Ntn1), an extracellular molecule expressed in the spinal cord (*Serafini et al., 1996*; *Figure 1—figure supplement 1A–B*). Early studies of *Ntn1* deletion based on a gene-trap allele (*Ntn1$^{\beta/\beta}$*) as well as targeted deletion alleles have identified aberrant entry of sensory axons into the dorsal spinal cord at E11.5 and E12.5 (*Bin et al., 2015*; *Varadarajan and Butler, 2017*; *Varadarajan et al., 2017*; *Wang, 1999*; *Watanabe et al., 2006*; *Wu et al., 2019*; *Yung et al., 2015*). Because of its expression immediately adjacent to the DREZ at E11.5 (*Serafini et al., 1996*), it has been suggested that Ntn1 prevents premature ingrowth of sensory afferents into the spinal cord via repulsion, an activity that was demonstrated in vitro (*Masuda et al., 2008*; *Watanabe et al., 2006*). However, it is not clear whether Ntn1 is also required for proper DRG axon bifurcation, and if so, whether Ntn1 is needed for forming the second branch or guiding bifurcated axons.

Using wholemount immunostaining coupled with tissue clearing (*Huber et al., 2005*; *Renier et al., 2014*; *Susaki et al., 2014*; *Tainaka et al., 2014*), we examined *Ntn1$^{\beta/\beta}$* mutants and found an ingrowth defect at the time of bifurcation, which differs in time and location from what has been previously reported (*Masuda et al., 2008*; *Watanabe et al., 2006*). Genetic and dye labeling confirmed that the misprojections were from DRG axons and single-cell analysis demonstrated that the defect resulted from misguidance of otherwise normally bifurcated axons. The misprojections had different trajectories from those found in *Slit1;Slit2* mutants (*Ma and Tessier-Lavigne, 2007*), and triple deletion of both pathways led to a rare and more severe phenotype with a near loss of the dorsal funiculus. Finally, a similar phenotype was also seen in mice lacking Ntn1 and Slit receptors. Taken together, these results demonstrate a new role for Ntn1, and more importantly, the presence of multiple guidance

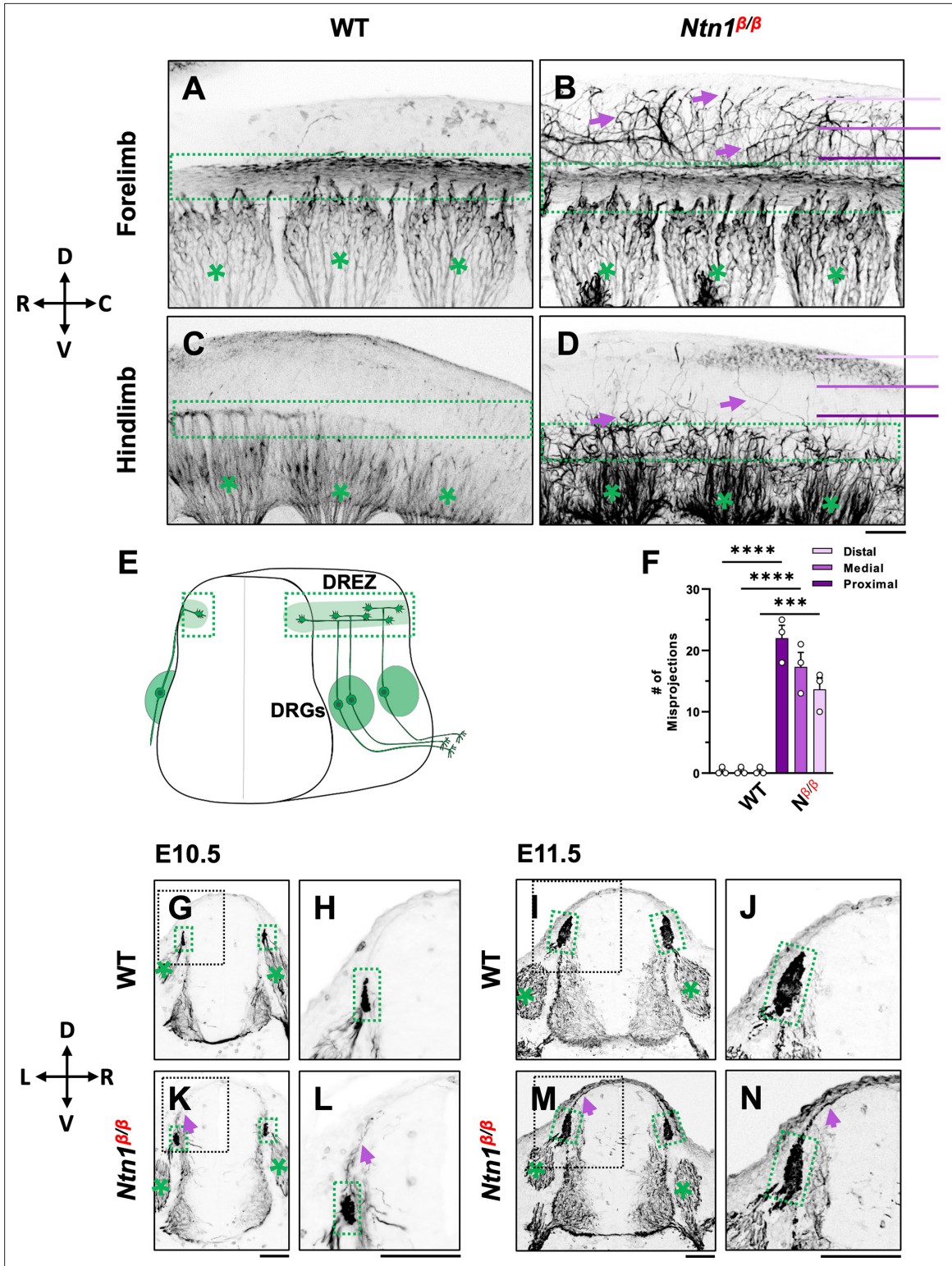

**Figure 1.** NF immunostaining reveals DRG axon misprojections in *Ntn1β/β* mutants in cleared wholemount embryos and transverse sections. (**A–D**) Inverted fluorescent images of wholemount NF staining of CUBIC-cleared E10.5 WT (**A, C**) or *Ntn1β/β* (**B, D**) embryos. The projected images are viewed from the lateral side of the body at the region of forelimb (**A–B**) and hindlimb (**C–D**). The space above the dorsal funiculus (dotted-line box) is devoid of NF-fibers in WT embryos (**A, C**), but filled with misprojecting axons (arrows) from DRGs (*) that grow dorsally in the mutant (**B,D**). The compass indicates the dorsal (**D**)-ventral (**V**) axis and the rostral (**R**)-caudal (**C**) axis. (**E**) A model depicting DRG sensory axon projections in the spinal cord. Bifurcation is shown in the DREZ (green area) that forms the eventual dorsal funiculus. (**F**) Quantification of the number of axonal fibers at the proximal,

*Figure 1 continued on next page*

*Figure 1 continued*

medial, or distal location from the dorsal funiculus (see purple lines in **B** and **D**) in the wholemount staining of WT and *Ntn1^(β/β)^* (*N^(β/β)^*) embryos (n=3 animals per condition). Two-way ANOVA F(1, 12)=199.5 with Tukey's HSD Test with multiple comparisons t between WT and *Ntn1^(β/β)^* i(proximal p<0.0001; medial p<0.0001; and distal, p=0.001). (**G–N**) Inverted fluorescent images of NF-stained transverse sections of the brachial region of E10.5 (**G–H, K–L**) and E11.5 (**I–J, M–N**) embryos. Misprojections (arrows) extending from the DREZ (green box) are seen in *Ntn1^(β/β)^* (**K–N**) but not WT (**G–J**) animals. Asterisks (*) label the DRG. Images in **H, J, L, N** are zoomed-in views of the boxed regions around the DREZ in **G, I, K, M**, respectively. The compass indicates the dorsal (**D**)-ventral (**V**) axis and the left (**L**)-right (**R**) axis. *** p<0.001, **** p<0.0001. Bars: 100 μm.

The online version of this article includes the following source data and figure supplement(s) for figure 1:

**Source data 1.** Quantification of the number of misprojecting axonal fibers at proximal, medial, and distal locations from the dorsal funiculus of WT and *Ntn1^(β/β)^* (*N^(β/β)^*) embryos (*Figure 1F*).

**Figure supplement 1.** *Ntn1* expression in mouse embryonic spinal cords and additional wholemount staining in E10.5 embryos.

mechanisms that confine the bifurcating axons within the DREZ and ensure the precise construction of the dorsal funiculus.

## Results

### Loss of Ntn1 causes axons to escape from the dorsal funiculus during bifurcation

To determine the role of Ntn1 in early DRG axon development, we first confirmed the expression of *Ntn1* in the spinal cord at E10.5, when DRG axons start to bifurcate and form the dorsal funiculus (*Ma and Tessier-Lavigne, 2007*; *Ozaki and Snider, 1997*). Consistent with previous published expression data (*Serafini et al., 1994*), *Ntn1* transcripts are found mainly confined to the inside of the spinal cord, including the floor plate, the ventricular zone, and the lateral domain in the dorsal horn below the dorsal funiculus along the pial layer from E10.5 to E11.5 (*Figure 1—figure supplement 1A–B* arrows). This RNA expression profile is in line with the localization of Ntn1 proteins made by ventricular zone neural progenitor cells and deposited along the pial layer adjacent to the DREZ (*Dominici et al., 2017*; *Varadarajan and Butler, 2017*).

To examine the DRG axonal tracts as they are entering the spinal cord at the DREZ, we performed wholemount neurofilament (NF) antibody staining combined with the CUBIC tissue clearing method (*Susaki et al., 2014*; *Tainaka et al., 2014*). Individual DRGs and their central projections in the dorsal funiculus can be seen from the lateral side of the embryos by confocal microscopy. As shown in wild type (WT) embryos, the central DRG axons first extend dorsally, enter the spinal cord, bundle and form the dorsal funiculus that runs along the rostrocaudal axis (*Figure 1A and C*). Due to the developmental delay along the rostrocaudal axis, the dorsal funiculus shown as a longitudinal track is thicker near the forelimb (*Figure 1A*) than near the hindlimb (*Figure 1C*), where it has just begun to form from bifurcated axons. These labeled axons stay within the tract forming tightly bundled fascicles, with few labeled axons extending dorsally from the dorsal funiculus (*Figure 1A*). This clear projection pattern can be also seen by traditional NF staining using horseradish peroxidase (HRP)-based immunostaining and BA/BB-based tissue clearing (*Huber et al., 2005*; *Ma and Tessier-Lavigne, 2007*; *Figure 1—figure supplement 1*). When viewed on a stereoscope, the labeled DRG axons appear to stay within the tract, leaving a clear space between left and right dorsal funiculi (*Figure 1—figure supplement 1C, D, G and H*).

Similar to those of WT embryos, DRG projections in *Ntn1^(β/β)^* mutants also form the dorsal funiculus, which can be seen as a NF-labeled longitudinal tract in the forelimb region (*Figure 1B*). However, while some axons remain in the tract, a subpopulation of NF-labeled fibers (*Figure 1B* arrows) were found to escape the tract, filling the space above the dorsal funiculus. These misprojecting fiber can be seen in traditional NF staining (*Figure 1—figure supplement 1E, F, I, J*). They appear in the hindlimb region where the dorsal funiculus starts to emerge (*Figure 1D*), suggesting that the defect happens at the time of bifurcation.

To quantify the misprojections, we used line scans to determine the number of misprojecting fibers straying away from the dorsal funiculus of a single DRG in the fluorescent wholemount images viewed from the lateral side of the embryo. Lines were placed in three positions - proximal, medial, and distal - that are 75 μm apart between the DREZ and the roof plate (*Figure 1B*). In WT embryos, almost no

fibers were found at any of the three positions, indicating that axons remain inside the dorsal funiculus. In *Ntn1^{β/β}* mutants, an average of 22 fibers per DRG were found in the line proximal to the DREZ, 17 fibers/DRG were found in the medial line, and 14 fibers/DRG were found in the distal line that is closest to the midline (*Figure 1F*). The presence of NF-fibers in the dorsal midline of the *Ntn1*mutants suggests that misprojections climb up in the dorsal spinal cord.

To better understand the location of misprojections, we examined transverse sections of E10.5 (*Figure 1G, H, K, L*) and E11.5 (*Figure 1I, J, M, N*) embryos stained for NF. In the dorsal spinal cord of WT embryos, NF labels axons that extend from the DRG into the DREZ (*Figure 1G, H, I, J*). The DREZ is smaller at E10.5 than E11.5 (*Figure 1G and H* vs I,J), but at both ages, the intense NF staining that represents the cross-sections of the dorsal funiculus remains inside the DREZ. However, in *Ntn1^{β/β}* mutants, NF-labeled fibers extended out from the DREZ at both ages. More misprojections were found at E11.5, but they all grow along the pia surface (*Figure 1K, L, M and N*), consistent with the dorsally extending trajectories seen in the wholemount staining above. Thus, loss of Ntn1 caused NF-labeled axons to escape from the DREZ.

## Ntn1 is invovled in DRG branch guidance at the time of bifurcation

Since NF can label axons from different neuronal populations including DRG sensory neurons, commissural neurons, and other spinal cord interneurons (*Bin et al., 2015*; *Moreno-Bravo et al., 2019*; *Yung et al., 2015*), we wanted to confirm that the observed misprojections at E10.5 are indeed axons from DRG sensory neurons. We thus introduced a neural specific CreER^{T2} recombinase driver (*Neurog1-CreER^{T2}*) and an Ai14 Cre reporter to our *Ntn1* mice to label all DRG neurons (*Koundakjian et al., 2007*). After Tamoxifen treatment to activate the Cre recombinase, we used RFP antibody to label the fluorescent protein tdTomato, which is expressed from the Ai14 allele, and examined DRG neurons and their axons in iDISCO-cleared and antibody-stained wholemount embryos (*Renier et al., 2014*).

In WT control embryos, tdTomato labeling is seen in DRG cell bodies (*Figure 2A*, asterisks), central axons projecting toward the dorsal spinal cord, and the dorsal funiculus (*Figure 2A*, yellow dotted box), which are also labeled by NF staining (*Figure 2A'*). In *Ntn1^{β/β}* mutant embryos, tdTomato labeling is seen in DRG central projections as well misprojections that escape from the DREZ and reach the dorsal spinal cord (*Figure 2B*). NF staining similar to *Figure 1* shows the misprojecting fibers (*Figure 2B'*) invading the dorsal space. While not all NF misprojections are tdTomato positive, all tdTomato-labeled misprojections are NF-positive (*Figure 2B"*). This can be demonstrated by closer examination of the region above the DREZ (*Figure 2C*) where most misprojecting fibers are stained for both tdTomato and NF (*Figure 2C"*, yellow arrows) but only a few are stained for just NF alone (*Figure 2C' and C"*, white arrow). Importantly, these tdTomato-labeled misprojections came directly from the DREZ (*Figure 2C*). D2 dorsal interneurons inside the spinal cord are known to express *Neurog1* (*Helms and Johnson, 2003*; *Koundakjian et al., 2007*). Although they are labeled by tdTomato staining, D2 interneurons are not labeled by NF (see *Figure 2A and A"*) and thus do not contribute to the NF-labeled misprojections. On the other hand, since *Neurog1-CreER^{T2}* may not be active in all DRG neurons, tdTomato-labeled misprojections likely underrepresent the DRG axon misprojections. Thus, these data support that some if not all NF-labeled misprojections seen above are from DRG sensory neurons.

To further confirm the DRG origin of misprojecting fibers found in *Ntn1* mutants and examine the nature of the defect, we next investigated the misprojections at the single axon level. We utilized the leaky recombinase activity of CreER^{T2} in the absence of tamoxifen, which leads to tdTomato expression, seemingly at random in a small population of DRG neurons and thus sparsely labels their axons. In iDISCO cleared E10.5 WT embryos, tdTomato labeling revealed that DRG axons displayed two types of morphologies in the DREZ: (1) bifurcated axons (*Figure 2D*), which have already reached the DREZ and bifurcated to extend the two branches in the dorsal funiculus that can be visualized by NF co-staining (*Figure 2F*); (2) non-bifurcated axons (*Figure 2E*), which have not yet bifurcated but reached the spinal cord and turned into the DREZ. Both bifurcated and non-bifurcated axons (n=12 vs 14) stay within the DREZ (*Figure 2I*) in the WT spinal cord. Similar bifurcated (n=17) and non-bifurcated (n=13) axons are found in *Ntn1^{β/β}* mutants (*Figure 2I*). However, although some bifurcated axons exhibit branches that correctly turn at the DREZ and follow the dorsal funiculus (*Figure 2G*, left branch), others often project dorsally out of the DREZ, indicating guidance errors (*Figure 2G*, right branch). Non-bifurcated axons often enter the DREZ and project away from the dorsal funiculus

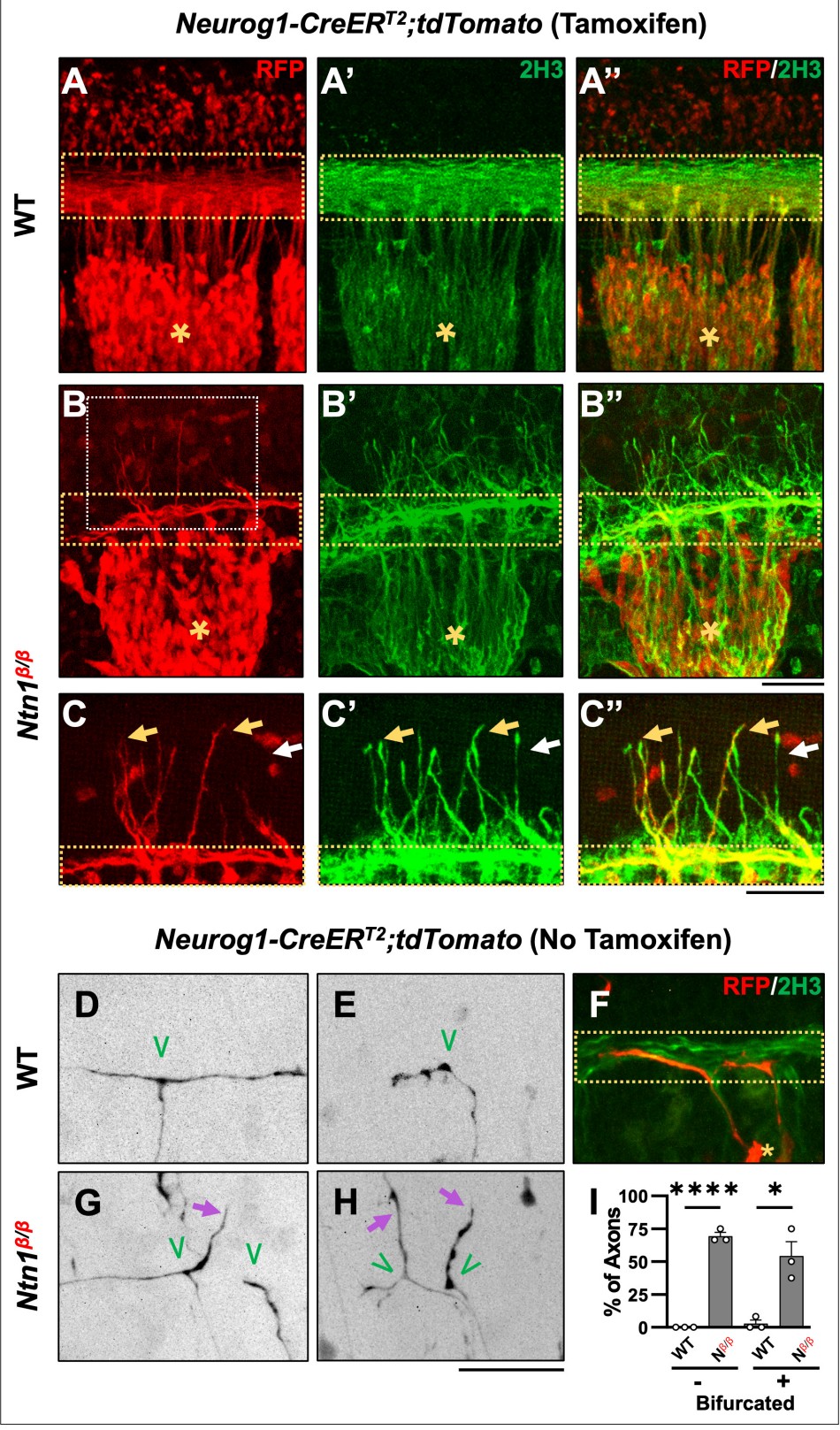

**Figure 2.** Genetic labeling of DRG neurons confirms the origin of misprojections and reveals guidance defects during bifurcation in *Ntn1^{β/β}* mutants. (A–C) Sensory axon specific genetic labeling based on tdTomato expression driven by tamoxifen-induced *Neurog1:CreER^{T2}* in E10.5 WT (A) or *Ntn1^{β/β}* (B) embryos that were cleared by iDISCO. Fluorescent images of immunostaining using RFP antibodies (for tdTomato) (A–C) or 2H3 antibodies (for NF)

*Figure 2 continued on next page*

*Figure 2 continued*

(**A'-C'**) are taken from the lateral side of the spinal cord in the brachial region of cleared wholemount embryos. Color-merged images are shown in **A"-C"**. Dorsal funiculi are inside the yellow dotted-line boxes and DRGs are marked by asterisks (*). Zoomed-in images (**C**) from the boxed regions in **B** (white box) highlights the co-staining of DRG-specific tdTomato and NF in abnormal fibers (yellow arrows) that extend from the dorsal funiculus in $Ntn1^{\beta/\beta}$ mutants (n=3). White arrows indicate NF-labeled misprojections that do not co-stain for tdTomato. (**D–H**) Sparse sensory axon labeling in WT or $Ntn1^{\beta/\beta}$ animals based on the basal tdTomato fluorescence driven by $Neurog1$-$CreER^{T2}$ without using tamoxifen. Inverted fluorescent images (**D,E,G,H**) are viewed along the DREZ from the lateral side of the spinal cord in cleared wholemount embryos. The merged fluorescent image of tdTomato and NF labeling shows the DREZ (dotted-line box) in **F**. Normal turning is found in WT animals (n=3) for bifurcated (**D**, n=11/12 axons) and non-bifurcated axons (**E**, n=14/14 axons). In $Ntn1^{\beta/\beta}$ mutant embryos (n=3), tdTomato labeled axons (purple arrows) can be seen misprojecting from the DREZ in bifurcated (**G,H**, n=7/14 axons) and non-bifurcated axons (**H**, n=9/13 axons). Green 'V' indicates the location of bifurcation junctions or the turning point of single axons. (**I**) Quantification of the defects in sparsely labeled axons above. The percentage of fibers that correctly (black bar) or incorrectly (grey bar) turn at the DREZ is quantified in two groups: bifurcated (+) or non-bifurcated (-). Student's t-tests compare the mean value of percentage of fibers misprojecting between WT and $Ntn1^{\beta/\beta}$ ($N^{\beta/\beta}$) in non-bifurcated axons (t(4)=25, p<0.0001) and bifurcated axons (t(4)=4.533, p=0.0106). * p<0.05, and **** p<0.0001, Bars: 100 μm.

The online version of this article includes the following source data for figure 2:

**Source data 1.** Quantification of the percentage of defects of sparsely labeled axons in WT and $N^{\beta/\beta}$ embryos (*Figure 2I*).

---

(*Figure 2H*). For comparison, nearly all axonal terminals stayed within the DREZ of WT embryos (*Figure 2I*), whereas ~55% bifurcated and ~70% non-bifurcated axons in the $Ntn1^{\beta/\beta}$ mutants misprojected outside of the defined dorsal funiculus (*Figure 2I*). These results demonstrate that loss of $Ntn1$ alters the guidance of DRG axons at the DREZ during bifurcation.

## Ntn1 and Slit regulate different aspects of sensory axon guidance during bifurcation

Our previous studies showed that impaired Slit signaling led to misguidance of one of the bifurcated DRG branches, which enter the dorsal spinal cord in the absence of Slit/Robo signaling (*Ma and Tessier-Lavigne, 2007*). Since the misprojections in $Ntn1$ mutants appear on the dorsal pial surface and stay outside the spinal cord, we hypothesize that Ntn1 and Slit have different guidance functions that are both required for the formation of the dorsal funiculus. To test this hypothesis, we generated triple mutants ($Slit1^{-/-};Slit2^{-/-};Ntn1^{\beta/\beta}$) and examined sensory axons in wholemount embryos using NF staining and tissue clearing as described in *Figure 1*. Triple mutants were generated in the $Slit1$ null background ($Slit1^{-/-};Slit2^{+/+};Ntn1^{+/+}$), which has nearly normal DRG axon projections (*Figure 3A*), similar to those in the WT embryos described above (*Figure 1A and C*). Once reaching the DREZ, axons form compact bundles that run along the rostrocaudal axis. A few short stray axons, due to the loss of $Slit1$, enter the normally clear space between the dorsal funiculi of the two sides of the embryo (*Figure 3A*). In contrast, NF labeled axons are completely disorganized around the DREZ (dotted-line region) in triple mutants ($Slit1^{-/-};Slit2^{-/-};Ntn1^{\beta/\beta}$) (*Figure 3B*). DRG axons still grow dorsally extending from the cell body, but rarely turn into the longitudinal track of the DREZ. Most of them continue to grow straight (*Figure 3B*, arrows) or veer off slightly after passing the presumptive dorsal funiculus. Examination using traditional DAB staining described earlier also confirms this complete ablation of the funiculus in triple mutants (*Figure 3—figure supplement 1A–D*) when viewed in either the forelimb or the hindlimb region. This analysis of triple mutants demonstrates that both Slit and Ntn1 are needed for the formation and integrity of the dorsal funiculus and supports the notion that they have different guidance roles during DRG axon bifurcation.

To gain further insights and distinguish the role of Slit and Ntn1, we next compared the trajectory and location of NF-labeled misprojections in embryos lacking $Slit1$ only, both $Slit1$ and $Slit2$, both $Slit1$ and $Ntn1$, or all three genes by visualizing them from different angles. First, when viewed from the lateral side of the wholemount embryos as above (*Figure 3C*), $Slit1$ single mutants ($Slit1^{-/-};Slit2^{+/+};Ntn1^{+/+}$) (*Figure 3D*) have a compact DREZ with few DRG axons projecting away. In double mutants lacking both Slit1 and Slit2 ($Slit1^{-/-};Slit2^{-/-};Ntn1^{+/+}$), an increased number of DRG axons leave the DREZ (*Figure 3E*, arrows) and appear in the proximal region of the dorsal spinal cord (*Figure 3E*),

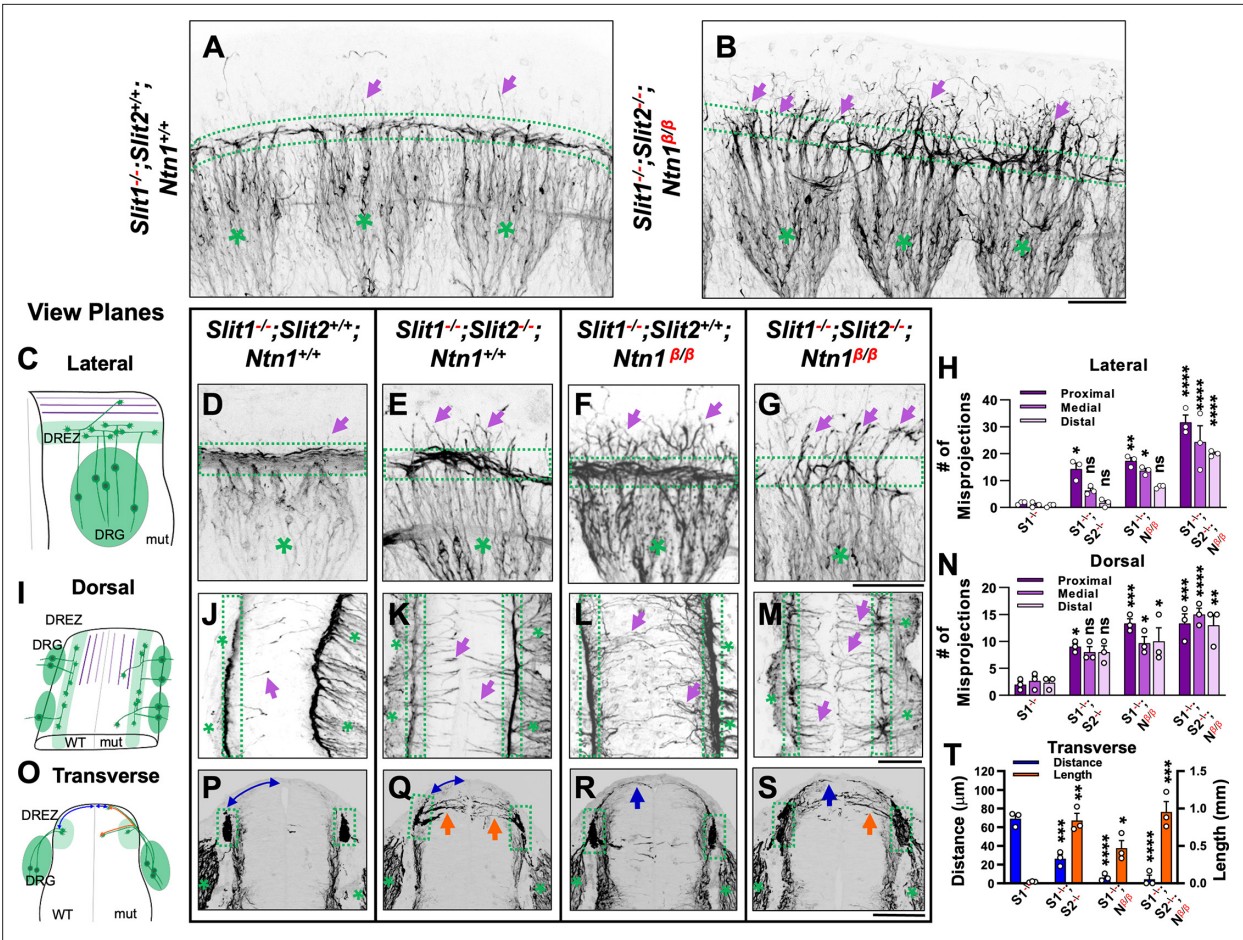

**Figure 3.** Wholemount NF immunostaining reveals the loss of the dorsal funiculus in *Slit1⁻/⁻;Slit2⁻/⁻;Ntn1^β/β* triple mutants with distinct misprojections. (**A–B**) Inverted fluorescent images of NF staining in CUBIC-cleared E10.5 embryos in the forelimb region. The dorsal funiculus (between dotted-lines) is evident in the control *Slit1⁻/⁻;Slit2⁺/⁺;Ntn1⁺/⁺* embryo and the space above it has few labeled fibers (arrows, **A**), but in the *Slit1⁻/⁻;Slit2⁻/⁻; Ntn1^β/β* mutant (**B**), extensive NF-labeled axons (arrows) wander into the dorsomedial region of the spinal cord while the dorsal funiculus (between dotted-lines) is reduced and diminished. DRGs are marked by green asterisks (*). (**C–N**) Inverted fluorescent images of NF staining in wholemount embryos of E10.5 embryos with various genotypes are viewed from the lateral side (**D–G**) or the dorsal surface (**J–M**). Arrows denote sensory misprojections found in *Slit1⁻/⁻;Slit2⁺/⁺;Ntn1⁺/⁺* (**D, J**), *Slit1⁻/⁻;Slit2⁻/⁻;Ntn1⁺/⁺* (**E, K**), *Slit1⁻/⁻;Slit2⁺/⁺; Ntn1^β/β* (**F, L**) and *Slit1⁻/⁻;Slit2⁻/⁻;Ntn1^β/β* (**G, M**) animals. In *Slit1⁻/⁻;Slit2⁻/⁻;Ntn1^β/β* triple mutants (**G, M**), the dorsal funiculus is nearly absent with misprojecting fibers. The number of axonal fibers at proximal, medial, and distal locations between the DREZ and the dorsal midline is quantified in the bar graphs from the lateral (**H**) or the top-down (**N**) view (n=3 for each data set). The orientation and measurement location are shown in the corresponding cartoons (**C and I**) and DRGs are marked by green asterisks (*). Statistics from two-way ANOVA analysis and pairwise comparions in **H**: for proximal misprojections, *Slit1⁻/⁻* vs. *Slit1⁻/⁻;Slit2⁻/⁻* p=0.0122; *Slit1⁻/⁻* vs. *Ntn1^β/β* p=0.0011; *Slit1⁻/⁻* vs. *Slit1⁻/⁻;Slit2⁻/⁻;Ntn1^β/β* p<0.001; for medial misprojections, *Slit1⁻/⁻* vs. *Ntn1^β/β* p=0.0122; *Slit1⁻/⁻* vs. *Slit1⁻/⁻;Slit2⁻/⁻;Ntn1^β/β* p<0.0001; and for distal misprojections, *Slit1⁻/⁻* vs. *Slit1⁻/⁻;Slit2⁻/⁻;Ntn1^β/β* p<0.0001. Statistics from two-way ANOVA analysis in **I**: for proximal misprojections, *Slit1⁻/⁻* vs. *Slit1⁻/⁻;Slit2⁻/⁻* p=0.0388; *Slit1⁻/⁻* vs. *Ntn1^β/β* p=0.0002; *Slit1⁻/⁻* vs. *Slit1⁻/⁻;Slit2⁻/⁻;Ntn1^β/β* p<0.0002; for medial misprojections, *Slit1⁻/⁻* vs. *Ntn1^β/β* p=0.0388; *Slit1⁻/⁻* vs. *Slit1⁻/⁻;Slit2⁻/⁻;Ntn1^β/β* p<0.0001; and for distal misprojections, *Slit1⁻/⁻* vs. *Ntn1^β/β* p=0.0175; *Slit1⁻/⁻* vs. *Slit1⁻/⁻;Slit2⁻/⁻;Ntn1^β/β* p=0.0004. (**O–T**) Inverted fluorescent images of NF staining in cross sections of E10.5 embryos with various genotypes (**P-S**, n=3). Orange arrows indicate axonal misprojections found in *Slit1⁻/⁻;Slit2⁻/⁻* mutants inside the spinal cord and blue arrows indicate misprojections associated with *Ntn1^β/β* mutants at the pial surface. Blue lines with arrow at both ends indicate the distance from dorsal projections to the midline. Quantification of defects (**T**) is based on the distance of the dorsal fibers from the midline (blue bars) and the total length of misprojection fibers inside the spinal cord (orange bars). The cartoon (**O**) illustrates the two types of misprojections (brown lines) and the quantification of dorsal distance (blue lines). Statistics from one-way ANOVA analysis in **T**: for dorsal fiber distance, WT vs. *Slit1⁻/⁻;Slit2⁻/⁻* p=0.0003; WT vs. *Ntn1^β/β* p<0.0001; WT vs. *Slit1⁻/⁻;Slit2⁻/⁻;Ntn1^β/β* p<0.0001; and for horizontal fiber length, WT vs. *Slit1⁻/⁻;Slit2⁻/⁻* p=0.0016; WT vs. *Ntn1^β/β* p=0.0472; WT vs. *Slit1⁻/⁻;Slit2⁻/⁻;Ntn1^β/β* p=0.007. * p<0.05, ** p<0.01, *** p<0.001 and **** p<0.0001, ns, not significant. Bars: 100 µm.

The online version of this article includes the following source data and figure supplement(s) for figure 3:

**Source data 1.** Quantification of the number of misprojecting axons identified from lateral (*Figure 3H*) or dorsal (*Figure 3N*) side of the spinal cord as well as the distance between the dorsal fiber and the midline and the total length of misprojecting axons on the transverse section (*Figure 3T*) of the spinal cord from animals with different genotypes.

**Figure supplement 1.** Traditional wholemount NF immunostaining reveals the loss of the dorsal funiculus in *Slit1⁻/⁻;Slit2⁻/⁻;Ntn1^β/β* triple mutants.

similar to those reported previously (*Ma and Tessier-Lavigne, 2007*). The number of misprojections tapers off more distally from the DREZ (*Figure 3E*). This can be demonstrated by viewing the misprojections from the dorsal side of the spinal cord (*Figure 3K*), as those misprojections maintain a relatively straight trajectory perpendicular to the DREZ. They also reach the midline but stop there without crossing to the contralateral side of the spinal cord. Such a trajectory is consistent with their invasion into the spinal cord, which can be shown on the transverse-section of the spinal cord. There, NF-labeled fibers appear to enter the spinal cord from the DREZ with horizontal trajectories inside the dorsal proper (*Figure 3Q*, arrows). These misprojections reach the midline with an average length of 137.8 µm and a summed length of ~0.8 mm, indicating a high number of misprojections (*Figure 3T*). For comparison, few misprojections are found in the *Slit1^{-/-}* animal on transverse sections, with a summed length of only 0.02 mm (*Figure 3T*).

As in *Ntn1* mutants, *Slit1^{-/-};Slit2^{+/+};Ntn1^{β/β}* embryos have a large number of dorsal misprojections that spread out in random directions from the DREZ along the dorsal spinal cord when viewed from the side (*Figure 3F*, arrowheads). They appear to extend away from the DREZ and reach more distal edge of the spinal cord (*Figure 3F and H*), different from that in *Slit1^{-/-};Slit2^{-/-};Ntn1^{+/+}* mutants. This can be corroborated by the dorsal view, which demonstrates that misprojections are wavy and long, sometimes passing the midline and reaching to the contralateral side of the spinal cord (*Figure 3L*). Such pattern is the result of misprojections emerging from the DREZ and growing upward along the pial layer as shown on the transverse section (*Figure 3R*). As a consequence, the distance from the midline to the DREZ is reduced to ~6 µm from ~69 µm of the *Slit1^{-/-}* control animal (*Figure 3T*). Although there are similar total numbers of misprojections as those in the *Slit1^{-/-};Slit2^{-/-};Ntn1^{+/+}* animals (*Figure 3T*), most of them stay outside the dorsal proper of the spinal cord (*Figure 3R*), indicating the different requirements of Ntn1 and Slit in the guidance of bifurcating DRG axons.

The independent roles are further demonstrated in triple mutants (*Slit1^{-/-};Slit2^{-/-};Ntn1^{β/β}*). In addition to the nearly complete loss of the dorsal funiculus, misprojecting fibers grow in either straight or in randomized directions, and increase their reach to the distal region (*Figure 3C and G*, arrows). As a result, the phenotype of triple mutants is more severe than that in *Ntn1^{β/β}* mutants (*Figure 1*, *Figure 3F*) or *Slit1;Slit2* double mutants (*Figure 3E*), which is demonstrated by an increase in these dorsally oriented fibers extending at all three proximal-medial-distal locations between the DREZ and the roof plate (*Figure 3H*). From the dorsal view, both straight and randomly oriented misprojections co-exist (*Figure 3M*), and from the cross section, misprojections are found both inside the dorsal spinal cord and along the pial surface, demonstrating additive guidance errors when both signaling pathways are impaired (*Figure 3H, N and T*). In addition, due to few axons turning into the DREZ, the distinct linear line in the DREZ appears uneven and often broken in the dorsal view (*Figure 3M*) and the DREZ became elongated in the transverse section (*Figure 3S*).

Taken together, these analyses from different views of the DREZ reveal distinct phenotypes in mice lacking Ntn1 and Slit1/2, thus supporting the different guidance roles of these pathways during dorsal funiculus development.

## Single axon analysis by DiI indicates neither Ntn1 nor Slit is required for branch formation

To determine whether Ntn1 is required for bifurcation, the process of forming the second branch of DRG sensory axons, we used DiI iontophoresis to label and examine single DRG axons at E12.5, the age when the majority of axons normally have already generated two branches (*Ma and Tessier-Lavigne, 2007*; *Ozaki and Snider, 1997*). Spinal cords were removed and imaged from an open book preparation (*Figure 4A*). Individuals axons were analyzed from maximum projections that were color coded for the depth in the z-plane (*Figure 4B and C*) Indeed, in WT embryos, labeled axons properly turn at the DREZ and 26 out of 27 axons have T-shaped junctions at the DREZ (*Figure 4B and D*). In *Ntn1^{β/β}* mutants, 32 out of 34 axons analyzed also have two branches (*Figure 4F*), indicating that branch formation itself is not affected in the absence of Ntn1. Similarly, 28 out of 29 axons have bifurcated in the *Slit1^{-/-};Slit2^{-/-}* mutant embryos (*Figure 4E*), while 20 out of 21 labeled axons in the triple *Slit1^{-/-};Slit2^{-/-}; Ntn1^{β/β}* mutants have two branches (*Figure 4C and G*). Since bifurcation remains nearly intact at this age for all these mutants, we conclude neither Ntn1 nor Slit is required for the formation of two branches.

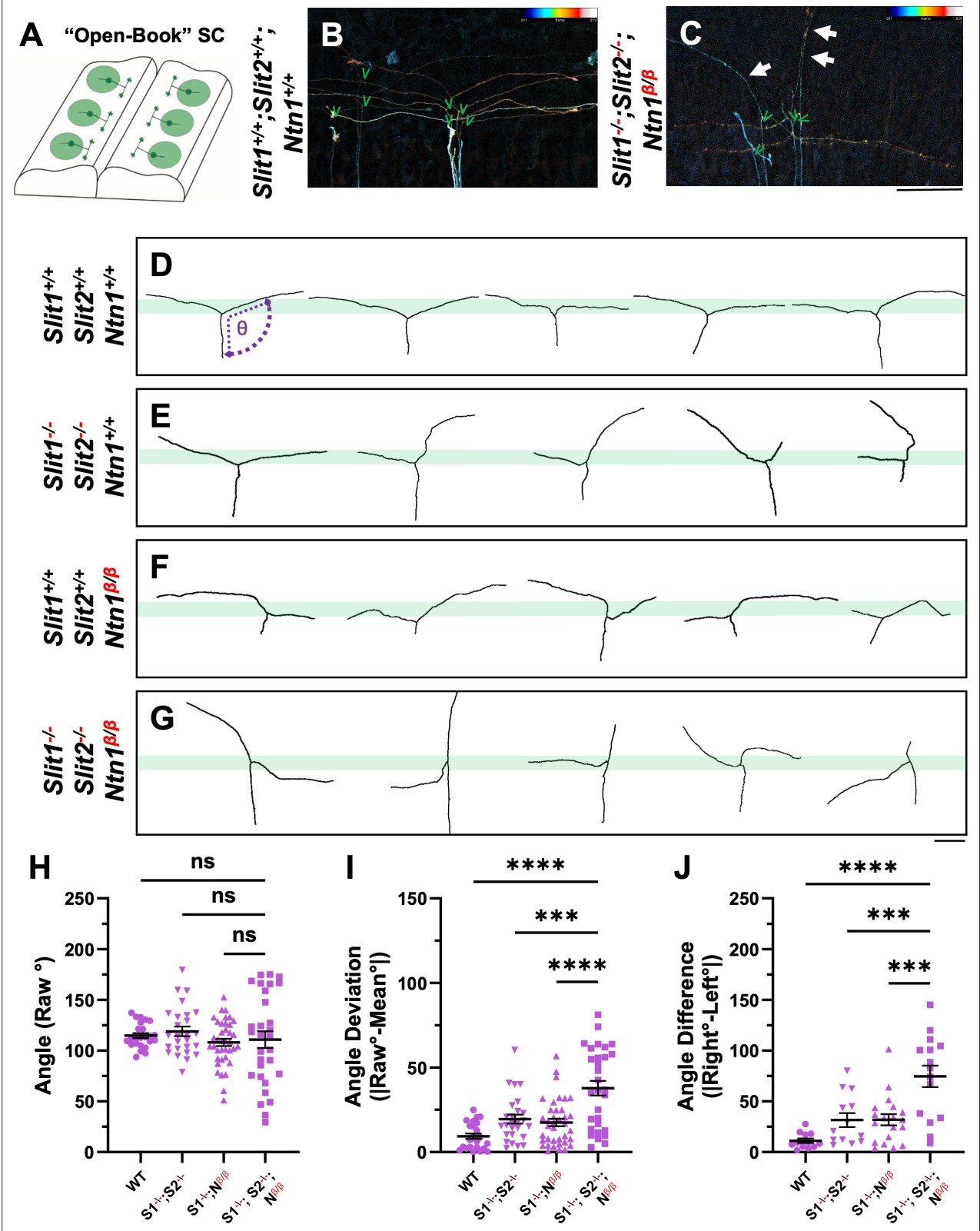

**Figure 4.** Single axon DiI labeling of DRG central afferents shows altered turning of bifurcated axons in various *Slit* and *Ntn1* mutants. (**A–C**) Single cell DiI labeling of DRG axons in open book preparations (**A**) of spinal cords isolated from E12.5 WT (**B**) and *Slit1⁻ᐟ⁻;Slit2⁻ᐟ⁻;Ntn1^β/β^* (**C**) animals. Maximum z-projection are depth color coded (~45 μm represented by the color bar) with multiple bifurcating axons visible in the field of view. Arrows point to misprojecting fibers and green "V" indicates the location of bifurcation junctions. (**D–G**) Skeletonized tracings of DiI labeled axons from embryos with

*Figure 4 continued*

various genotypes. WT axons show stereotyped T-bifurcation along the DREZ (**D**, n=4 animals). Axons in *Slit1⁻/⁻;Slit2⁻/⁻* mutants overshoot the DREZ (**E**, n=3 animals). In *Ntn1^{β/β}* mutant embryos, axons stray from the DREZ and have disrupted turns along the DREZ (**F**, n=5 animals). Axons in *Slit1⁻/⁻;Slit2⁻/⁻;Ntn1^{β/β}* triple mutants are completely disorganized around the DREZ (**G**, n=2 animals). Dashed lines in **D** show the angle ($\theta$) measurement. (**H–J**) Quantification of the turning angle of bifurcated branches along the DREZ (**H**) from single axon labeling above showed the mean raw angles are similar for all genotypes (one way ANOVA $F_{(3,114)}=0.8229$, p=n .s.). However, the deviation of raw angles from the average was significantly different among different genotypes (**I**). One way ANOVA with Tukey's Post-hoc for multiple comparisons ($[F_{(3,114)}=15.97$, p<0.0001]): WT and *Slit1⁻/⁻;Slit2⁻/⁻;Ntn1^{β/β}*, p<0.0001; *Slit1⁻/⁻;Slit2⁻/⁻* and *Slit1⁻/⁻;Slit2⁻/⁻;Ntn1^{β/β}*, p=0.0001; and *Ntn1^{β/β}* and *Slit1⁻/⁻;Slit2⁻/⁻;Ntn1^{β/β}*, p<0.0001. Additionally, the angle difference between the two branches showed significantly difference among genotypes (**J**). One way ANOVA with Tukey's Post-hoc for multiple comparisons ($[F_{(3,114)}=12.99$, p<0.0001]): WT and *Slit1⁻/⁻;Slit2⁻/⁻;Ntn1^{β/β}*, p<0.0001; *Slit1⁻/⁻;Slit2⁻/⁻* and *Slit1⁻/⁻;Slit2⁻/⁻;Ntn1^{β/β}*, p=0.0004; and *Ntn1^{β/β}* and *Slit1⁻/⁻;Slit2⁻/⁻;Ntn1^{β/β}*, p=0.0001. Only the comparison between the triple mutants and other genotypes are shown here. *** p<0.001, **** p<0.0001, ns: not significant. Bars: 100 μm.

The online version of this article includes the following source data for figure 4:

**Source data 1.** Analysis of the turning behavior of bifurcated branches along the DREZ based on raw angle (*Figure 4H*), angle deviation (*Figure 4I*), and angle difference (*Figure 4J*).

## Single axon analysis supports that Ntn1 and Slit are involved in different aspects of guidance at the DREZ

The DiI analysis also allowed us to analyze additional features of bifurcating DRG sensory axons. We examined the single axon trajectories by measuring and comparing two angles between the two branches and the primary axons (*Figure 4D*). In WT animals, the angle of bifurcation is on average 115.4° (*Figure 4H*), consistent with a more curved trajectory of bifurcation at this age. The average deviation is 10.2° (*Figure 4I*) and the difference between two angles from the same axons is 11.8° (*Figure 4J*), indicating similar projection patterns for the two bifurcated branches. In *Slit1⁻/⁻;Slit2⁻/⁻* mutant embryos (*Figure 4E*), some branches turn normally into the DREZ, while others often fail to properly turn but instead overshoot into the spinal cord, as portrayed by a relatively straight trace. When analyzed together, the average angle is 118.9°, only marginally larger than WT (*Figure 4H*), but the angle deviation is increased to 19.5° and the angle difference to 31.6° (*Figure 4I and J*), reflecting misguidance of one of the two branches (*Figure 4I*) and consistent with the previous finding (*Ma and Tessier-Lavigne, 2007*). A similar angle difference was found in *Ntn1^{β/β}* mutant embryos (*Figure 4F*). The average angle of bifurcation is ~108.0° (*Figure 4H*) with a deviation of 17.5° (*Figure 4I*) and a difference of 31.8°. However, analysis of *Slit1⁻/⁻;Slit2⁻/⁻; Ntn1^{β/β}* triple mutants (*Figure 4G*) reveals an enhanced defect with highly deviated trajectories. Although the average angle is 110.7°, similar to that of WT (*Figure 4H*), nearly all branches are deviated from the average angle by a margin of 37.8°, significantly different from that seen in either *Slit1;Slit2* or *Ntn1* mutants (*Figure 4I*). Some branches in the triple mutants even overshoot the DREZ with 180° angles, turn with a drastic kink, or have an altered axis of turning in the DREZ (*Figure 4G*). As a result, the angle difference is increased dramatically in the triple mutants, reaching 74.6° (*Figure 4J*). Taken together, these results demonstrate the additive roles of Ntn1 and Slit, suggesting that the guidance of the two branches has different requirements of Ntn1 and Slit.

## Role of DCC and Robo receptors in DRG axon guidance during dorsal funiculus formation

To further demonstrate the requirement of the two extracellular signals in guiding DRG sensory axons during bifurcation, we examined mice lacking the Ntn1 receptor DCC, which was previously shown to have a late-ingrowth defect (*Ding et al., 2005*). DCC is expressed at low levels in the DRG by in situ analysis (*Faure et al., 2020*; *Figure 5—figure supplement 1*), and recent analysis by single cell RNAseq suggests DCC is expressed in a subset of sensory neurons (*Faure et al., 2020*). Using wholemount analysis of E10.5 *Dcc* mutants, we found similar misprojections as those in *Ntn1* mutants (*Figure 5A and B*, arrows). Since the deletion of the Slit receptor Robo1 and Robo2 results in the same overshooting error as the loss of Slit1 and Slit2 (*Ma and Tessier-Lavigne, 2007*), we next generated a triple mutant lacking all three receptors (*Robo1⁻/⁻;Robo2⁻/⁻;Dcc⁻/⁻*). Fluorescent labeling of sensory axons in E10.5 wholemount embryos reveals extensive dorsal misprojections that fail to form axonal bundles of the dorsal funiculus in the triple mutant when compared with the littermate control (*Figure 5C and D*).

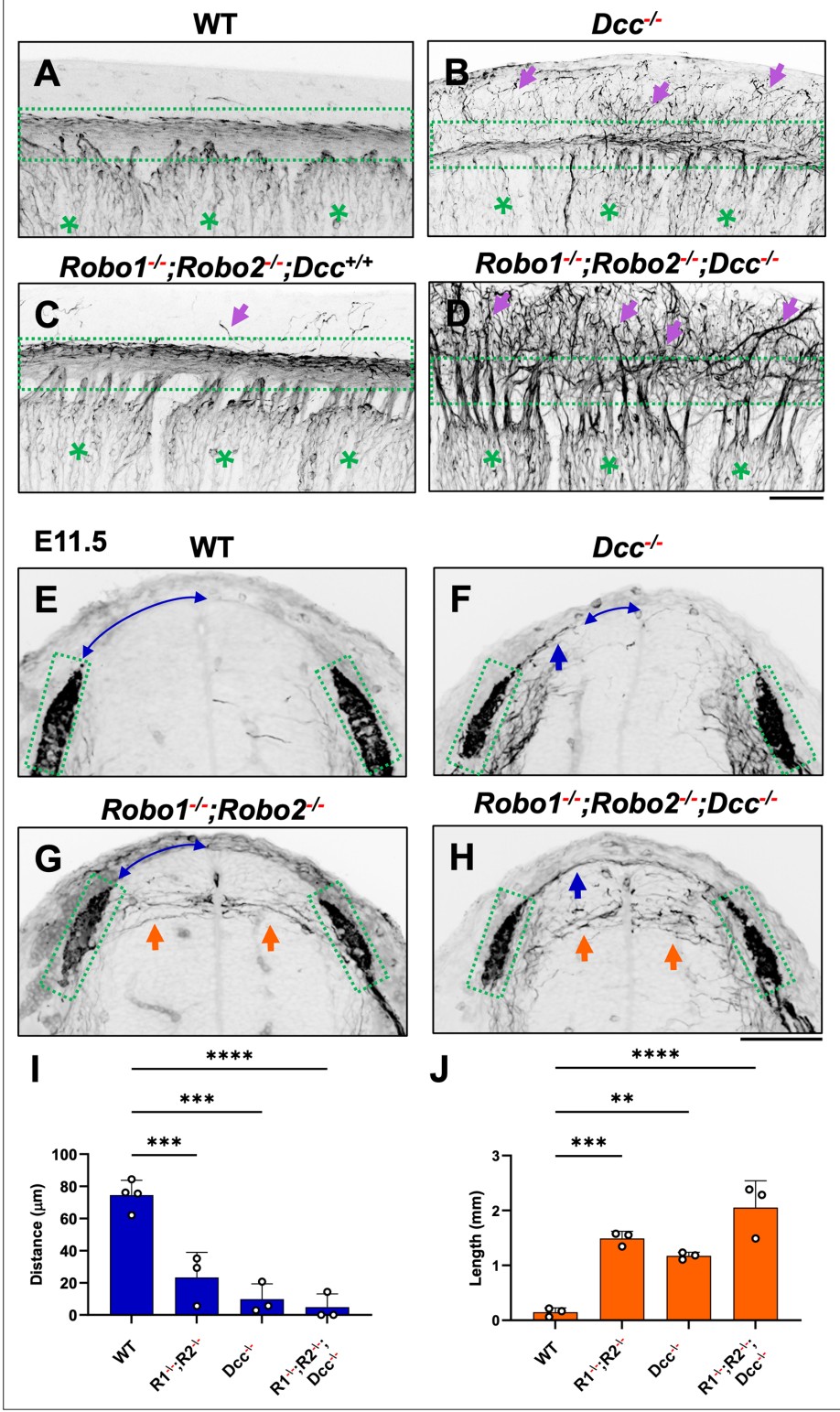

**Figure 5.** NF immunostaining reveals the loss of the dorsal funiculus in *Robo1^{−/−};Robo2^{−/−};Dcc^{−/−}* triple mutants. (**A–D**) Inverted fluorescent images of NF staining in CUBIC-cleared E10.5 embryos viewed from the lateral side in the forelimb region. WT (**A**, n=3) have relatively few axons straying from the dorsal funiculus (boxed region). Purple arrows denote sensory misprojections found in *Robo1^{−/−};Robo2^{−/−};Dcc^{+/+}* embryos (**C**, n=2) or *DCC^{−/−}* (**B**, n=6). In *Robo1^{−/−};Robo2^{−/−};Dcc^{−/−}* mutants (**D**, n=3), the dorsal funiculus is completely lost and disorganized with axons wandering into the dorsomedial region of the spinal cord. (**E–H**) Inverted fluorescent images of NF staining

*Figure 5 continued on next page*

*Figure 5 continued*

in the cross section of E11.5 embryos with various genotypes. Orange arrows indicate horizontal misprojections inside the spinal cord found in *Robo1⁻/⁻;Robo2⁻/⁻* mutants and blue arrows indicate dorsal misprojections along the pial surface associated with *Dcc⁻/⁻* mutants. Lines with arrow at both ends (blue) indicate the distance between dorsal misprojections to the midline. (**I–J**) Quantification of dorsal and horizontal misprojections based on the distance from the roof plate to the DREZ (**I**, blue bars, n=3, except n=4 for WT) and the total length of horizontal fibers (**J**, orange bars, n=3). Statistics from one-way ANOVA analysis: for dorsal fiber distance in **I**, WT vs. *R1⁻/⁻;R2⁻/⁻* p=0.0008; WT vs. *DCC⁻/⁻* p=0.0001; WT vs. *R1⁻/⁻;R2⁻/⁻; DCC⁻/⁻* p<0.0001; for dorsal fiber length in **J**, WT vs. *R1⁻/⁻;R2⁻/⁻* p=0.0010; WT vs. *DCC⁻/⁻* p=0.0054; WT vs. *R1⁻/⁻;R2⁻/⁻; DCC⁻/⁻* p<0.0001. ** p<0.01; *** p<0.001; **** p<0.0001, ns: not significant. Bars: 100 μm.

The online version of this article includes the following source data and figure supplement(s) for figure 5:

**Source data 1.** Quantification of the distance between the dorsal fiber and the roof plate (*Figure 5I*) and the total length of the misprojecting axons (*Figure 5J*) from the transverse section of the spinal cord of different genotypes.

**Figure supplement 1.** *Dcc* expression in mouse embryonic spinal cords.

To characterize the misprojections quantitatively, we examined E11.5 spinal cord transverse-sections and found the same two types of NF-labeled misprojections seen in mutants lacking Slit and Ntn1 above (*Figure 3*). In *Robo1⁻/⁻;Robo2⁻⁻* mutant embryos (*Figure 5G*), , the DREZ still forms but some fibers invade the spinal cord with straight trajectories that shoot horizontally in the dorsal spinal cord (*Figure 5I*), similar to that in *Slit1⁻/⁻;Slit2⁻/⁻* cross sections (*Figure 3*). In *Dcc⁻/⁻* only mutants, the DREZ still forms and the NF-labeled fibers misproject upward along the pial layer (*Figure 5F*), leading to a reduced distance between the midline and the DREZ (*Figure 5I*), similar to the *Ntn1* phenotype (*Figure 1L and M*). In *Robo1⁻/⁻;Robo2⁻/⁻;Dcc⁻/⁻* triple mutants, two types of misprojecting fibers are found, similar to those in *Slit1⁻/⁻;Slit2⁻/⁻;Ntn1^{β/β}* embryos (*Figure 3*). Both misprojections originate from the DREZ, but some stay dorsally on top of the pial surface (*Figure 5F and H*) while others enter the spinal cord horizontally (*Figure 5G and H*, arrows). When quantified, mice lacking either Robo1;Robo2 or DCC have similar total lengths of horizontal misprojections, but the triple mutants have increased total lengths, reflecting an increased number of misprojections (*Figure 5J*). In addition, the gap between the dorsal misprojections and the midline seems to be reduced significantly in all mutants (*Figure 5I*). These phenotypes are similar to those found in the analysis of *Slit1;Slit2* and *Ntn1* mutants above (*Figure 3*), thus demonstrating that loss of either the ligand or the receptor of both signaling systems has the same impact on DRG axon guidance at the DREZ.

## Discussion

Our in vivo analysis of two guidance pathways has identified multiple guidance mechanisms that shape DRG axons bifurcation when forming the dorsal funiculus in the DREZ. We show that development of this stereotyped T-shaped structure requires both Ntn1 and Slit, which regulate different aspects of guidance during bifurcation. These results demonstrate the presence of multiple mechanisms to ensure the proper formation of the dorsal funiculus, a structure that is essential to sensory function in the mammalian spinal cord (*Light, 1988*).

### Ntn1 signaling is required for guiding bifurcating DRG axons in the DREZ

Our analysis of mouse mutants lacking the secreted molecule Ntn1 expanded previous observation of a spinal cord ingrowth defect that was interpreted as the function of Ntn1 in a critical 'waiting period' before DRG axons sprout collateral branches (*Watanabe et al., 2006*). However, our study showed that loss of this molecular pathway also leads to a profound guidance error at the time of bifurcation, as misprojecting axons were found as early as E10.5, when the dorsal funiculus first forms (*Figure 1*). The misprojecting axons do not stop at the DREZ but instead project more dorsally along the pial surface with random paths, supporting the idea that Ntn1 is needed for proper guidance of DRG afferents right at the time when they bifurcate in the DREZ. The origin of the misprojections is confirmed to come from DRG neurons by both genetic labeling using *Neurog1-CreER^{T2}* (*Figure 2*) and DiI labeling (*Figure 4*). Single axon labeling by both methods further demonstrates the guidance role of Ntn1, which is different from what was suggested by previous studies (*Masuda et al., 2008*;

*Watanabe et al., 2006*). In these studies, DRG axons were found to enter from the ventral side of the DREZ in the medial spinal cord, leading to the proposal that Ntn1 blocks the DRG axons from entering the spinal cord when they initially arrive at the DREZ. Our data demonstrate that Ntn1 is required for guidance as well, but Ntn1 acts on bifurcating axons to guide their growth along the DREZ and loss of Ntn1 causes misprojections to exit the dorsal side of the DREZ (*Figure 1*). Moreover, single axon DiI labeling at a later stage rules out the requirement of Ntn1 in bifurcation as the majority of DRG axons in *Ntn1* mutants still form two branches (*Figure 4*). This is an important conclusion as premature entrance into the spinal cord could interfere or delay bifurcation, which was not analyzed previously (*Masuda et al., 2008*; *Watanabe et al., 2006*). Interestingly, both dorsal and medial misprojections were observed in the recent studies of the embryo with complete Ntrn1 deletion (*Moreno-Bravo et al., 2019*; *Varadarajan and Butler, 2017*; *Wu et al., 2019*), suggesting that Ntn1 may play multiple guidance roles. Regardless, our single axon analysis (*Figures 2 and 4*) demonstrates that Ntn1 exerts its guidance function on DRG axons at the time of bifurcation.

The guidance function is further supported by the similar defect found in mice lacking the Ntn1 receptor DCC (*Figure 5*). This DCC function is reminiscent to that found in zebrafish DRG axons, which require proper DCC signaling to control actin-mediated invadopodia during the initial entry into the spinal cord (*Kikel-Coury et al., 2021*). There are two possible mechanisms. One is that Ntn1 proteins present locally outside the DREZ (*Figure 1—figure supplement 1*; *Serafini et al., 1996*) provides a permissive cue that encourage the growth of newly bifurcated branches but restrict their trajectory within the DREZ, similar to the recently described role for Ntn1 in confining pontine neuron migration (*Yung et al., 2018*). Such a permissive function matches with the proposed adhesion function of DCC (*Meijers et al., 2020*). Alternatively, it is possible that the guidance function is mediated by a repulsive action of Ntn1, which can be mediated by both DCC and the repulsive receptor Unc5C. This is supported by Unc5C expression in DRG neurons as well as the repulsive activity of Ntn1 on E11.5 and E13.5 DRG axons in vitro (*Masuda et al., 2008*; *Watanabe et al., 2006*). Thus, it would be interesting to re-examine Unc5C mutants and determine its role during DRG axon bifurcation and distinguish these potential mechanisms for Ntn1 signaling. Regardless, our studies demonstrate a key role for Ntn1 signaling in DRG axon guidance right after bifurcation at the DREZ.

## Multiple mechanisms are present to regulate DRG axon guidance during bifurcation

One striking result from our studies is the complete disorganization and the loss of axonal bundles in the dorsal funiculus when both Slit and Ntn1 pathways are impaired. This is reflected by the severe sensory misprojections in triple mutants lacking all three genes, *Ntn1*, *Slit1*, and *Slit2*, in which nearly all DRG axons leave the DREZ and grow medially into the dorsal spinal cord (*Figure 3*). Similar problems were found in mice lacking the receptors for both pathways (*Figure 5*). To our knowledge, this is the first report of such a severe defect of the dorsal funiculus.

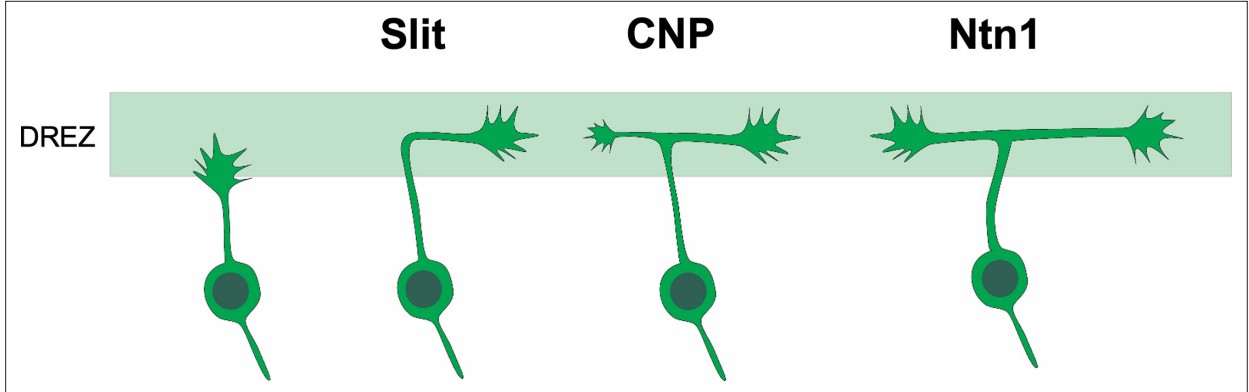

**Figure 6.** Cartoon illustration of the different guidance roles of Slit and Ntn1 during bifurcation. A working model for the role of Slit and Ntn1 in regulating branch guidance at two different steps during bifurcation. The primary axon of the central projection from DRG neurons first extends toward the dorsal spinal cord. After reaching the DREZ, the axon is turned by Slit into the rostrocaudal direction. After forming the second branch stimulated by CNP, the two bifurcated axons continue to grow along the DREZ with their trajectories regulated by a Ntn1-dependent guidance mechanism.

Importantly, when the misprojections were examined more carefully, we found that the guidance error found in *Ntn1* mutants is different from that found in mice lacking Slit (**Ma and Tessier-Lavigne, 2007**). The misprojections due to the loss of Slit signaling tend to enter the spinal cord horizontally whereas those from the loss of Ntn1/DCC grow more dorsally and stay at the pial surface (**Figure 3**). In addition, different growth trajectories were found for the misprojections: those in *Slit/Robo* mutants are relatively straight and horizontal whereas those in *Ntn1* mutants are wavy and randomly oriented. It is worth noting that the misprojections seen in *Slit/Robo* mutants enter the spinal cord from the middle or the dorsal side of the DREZ, which is different from those medial misprojections found in *Ntn1* mutants discussed above (**Moreno-Bravo et al., 2019**; **Varadarajan and Butler, 2017**; **Wu et al., 2019**). Nonetheless, these phenotypes suggest that Ntn1 and Slit have different actions in guiding DRG axons. This conclusion is further supported by the defects in triple mutants, which exhibit both types of misprojections. Quantitative analysis shows that loss of both pathways results in added defects for both horizontally and dorsally invading misprojections (**Figures 3–5**), suggesting non-redundant functions of the two mechanisms.

Building on a collateral-couple repulsion model we proposed earlier (**Gibson and Ma, 2011**), we suggest that Slit and Ntn1 regulate branch guidance at two different steps during bifurcation (**Figure 6**). When the primary axon of DRG neurons first reach the dorsal spinal cord, they are guided by Slit to turn into the DREZ; after turning, the second branch is stimulated to form by CNP; and finally, the two branches continue to grow along the DREZ and their trajectory is regulated by Ntn1. Future analysis at single-cell levels would be useful to test this refined model and distinguish the role of these molecules before and after bifurcation. Imaging the growth cone behaviors during bifurcation in vivo would provide additional information of the guidance decision as shown by recent studies in zebrafish and spinal cord explants (**Kikel-Coury et al., 2021**; **Nichols and Smith, 2019**; **Pignata et al., 2019**). Altogether, our new study demonstrates that formation of a simple bifurcation structure requires multiple molecular mechanisms to ensure not only the formation of the second branches but also the correct orientation of both branches.

## New insights of guidance regulation at the DREZ

The guidance function of Ntn1 and Slit has been well studied in midline guidance (**Chédotal, 2019**), but guidance regulation of DRG axons at the DREZ is less well understood. It is worth noting that midline crossing is defined by the floor plate, a structure that is formed during early spinal cord development and which serves as an intermediate target (**Chédotal, 2019**). However, in our view, the DREZ is an evolving structure that is not defined by any preexisting cell types. Rather, the DREZ is constructed at the location where DRG sensory axons first bifurcate, a process that involves the interaction of the DRG central axons with the cells in the dorsal spinal cord. This is best illustrated by the recent time-lapse imaging in the live zebrafish spinal cord, where the growth cone of pioneer DRG sensory neurons uses actin-rich filopodia and invadopodia to interact with the glial environment and brake axon growth at the future DREZ (**Kikel-Coury et al., 2021**; **Nichols and Smith, 2019**). Interestingly, DCC signaling was shown to control invadopodia stabilization during this precise guidance event, pointing to a critical role of the interaction between DRG axons and the local environment when entering the spinal cord (**Kikel-Coury et al., 2021**). Although our phenotype is consistent with this critical role, future live cell imaging is needed to examine how growth cones of rodent DRG axons are regulated to coordinate growth, guidance, and bifurcation during DREZ development. Nonetheless, our identification of the role of extracellular cues like Slit, Ntn1, and CNP (**Ma and Tessier-Lavigne, 2007**; **Zhao and Ma, 2009**) supports the hypothesis that both formation of the second branch and guidance of both bifurcated branches are critical to establishing the DREZ.

In addition to these extracellular cues, it has been suggested that sensory axon ingrowth and DREZ formation could be influenced by glial cells, especially boundary cap cells (**Golding et al., 1997**; **Golding and Cohen, 1997**; **Maro et al., 2004**). Since boundary cap cells are derived from late-born neural crest cells and associated with the dorsal root entry site starting at E12 in rat (**Golding and Cohen, 1997**), they could potentially shape the DREZ during initial DRG axon ingrowth and bifurcation. Although boundary cap cells do not seem to correlate with growth of pioneer DRG axons to the DREZ in the developing zebrafish spinal cord (**Nichols and Smith, 2019**), it would be interesting to examine their roles in rodent DRG axons by combining our single cell analysis with boundary cap cell-specific ablation (**Maro et al., 2004**). Besides boundary cap cells, dorsal interneurons, which

migrate ventrally at the time of DRG axon bifurcation, might also play a role here, as migration defects have been reported in both *Ntn1* and *Dcc* mutant embryos (*Ding et al., 2005*; *Junge et al., 2016*). These cells could produce extracellular factors to control DRG axon guidance. In fact, loss of TAG-1/ Axonin-1 from these cells cause similar but not identical afferent defects at the DREZ (*Perrin et al., 2001*). To determine whether Ntn1 acts directly on DRG axons or indirectly to influence DRG bifurcation via interneurons, cell-specific deletion (instead of global knockouts used in this study) is needed to distinguish cell autonomous vs non-cell autonomous roles of DCC in guiding DRG axons (*Suter and Jaworski, 2019*). Regardless of the mechanistic details, our in vivo analysis in various knockout mice provides the first step toward understanding the complex regulation of axonal development at the DREZ, the PNS-CNS interface that is critical to spinal cord function and is often disrupted in genetic disorders or after injuries (*Koeppen et al., 2017*; *Zheng et al., 2019*).

# Materials and methods

**Key resources table**

| Reagent type (species) or resource | Designation | Source or reference | Identifiers | Additional information |
|---|---|---|---|---|
| Strain, strain background (*Mus musculus*) | *Neurog1:CreER^{T2}* | The Jackson Laboratory | RRID:IMSR_JAX:008529 | A BAC transgenic mouse strain |
| Strain, strain background (*Mus musculus*) | Ai14 (tdTomato) | The Jackson Laboratory | RRID:IMSR_JAX:007914 | A knock-in mouse strain |
| Strain, strain background (*Mus musculus*) | *Ntn1^{β/β}* | *Serafini et al., 1996* | RRID:MMRRC_030660-UCD | A gene trap mouse strain |
| Strain, strain background (*Mus musculus*) | *Slit1;Slit2* | *Plump et al., 2002* | RRID:MMRRC_030404-MU | A knock-out mouse strain |
| Strain, strain background (*Mus musculus*) | *Dcc* | *Fazeli et al., 1997* | RRID:MMRRC_030626-MU | A knock-out mouse strain |
| Strain, strain background (*Mus musculus*) | *Robo1;Robo2* | *Ma and Tessier-Lavigne, 2007* | RRID:MMRRC_030747-MU | A knock-out mouse strain |
| Antibody | anti-neurofilament (Mouse monoclonal) | DSHB | Cat# 2H3, RRID:AB_531793 | IF(1:200) |
| Antibody | anti-RFP (Rabbit polyclonal) | Rockland | Cat# 600-401-379, RRID:AB_2209751 | IF(1:500) |
| Antibody | Cy5-AffiniPure Donkey Anti-Mouse IgG (H+L) | Jackson ImmunoResearch Labs | Cat# 715-175-150 RRID:AB_2340819 | IF(1:500) |
| Antibody | Cy3-AffiniPure Donkey Anti-Rabbit IgG (H+L) | Jackson ImmunoResearch Labs | Cat# 711-165-152, RRID:AB_2307443 | (IF1:500) |
| Other | DiI | Invitrogen | Cat# D282 | 0.5% in ethanol |
| Software, algorithm | PRISM 10 | GraphPad | RRID: SCR_002798 | https://www.graphpad.com/scientific-software/prism/ |
| Software, algorithm | Leica Application Suite X | Leica Microsystems | RRID:SCR_013673 | https://www.leica-microsystems.com/products/microscope-software/details/product/leica-las-x-ls/ |
| Software, algorithim | FIJI | *Schindelin et al., 2012* | RRID:SCR_002285 | https://fiji.sc |

## Mouse strains

All animal procedures followed the Guidelines for the Care and Use of Laboratory Animals of the National Institutes of Health and the approved IACUC protocols of the Thomas Jefferson University (#01558 and #01559) and of the Brown University (#21-12-0006). Mice were maintained in a CD-1 background. Timed pregnancies were determined based on vaginal plugs and the first date

was designated as E0.5. Mutant alleles for *Ntn1*, *Dcc*, *Slit1*, *Slit2* and *Robo1;Robo2* were described previously (*Fazeli et al., 1997*; *Ma and Tessier-Lavigne, 2007*; *Plump et al., 2002*; *Serafini et al., 1996*). The transgenic allele *Neurog1-CreER^T2* that express CreER^T2 recombinase from the Neurogenin-1 promoter was described (*Koundakjian et al., 2007*) and the Cre-reporter, Ai14(tdTomato), that expresses the red fluorescent protein tdTomato from a CAG promoter was described (*Madisen et al., 2010*). Homozygous *Ntn1^{β/β}* mutants were generated from heterozygous sires and dams. To obtain *Slit1^{-/-};Slit2^{-/-};Ntn1^{β/β}* triple mutants, mice carrying a single copy of *Slit2* and *Ntn1* mutant allele in the *Slit1^{-/-}* background were first generated and then used to obtain triple mutants. Genotyping of *Slit1*, *Slit2*, *Robo1*, *Robo2* and *Dcc* was done by PCR as previously described (*Fazeli et al., 1997*; *Ma and Tessier-Lavigne, 2007*; *Plump et al., 2002*); genotyping of Ai14(tdTomato) and *Neurog1:CreER^T2* was done by PCR using primers recomended by the Jackson Laboratory. For *Ntn1* genotyping, the intensity and kinetics of lacZ staining of embryonic tails were used to determine the copy number of the mutant allele and the resulting *Ntn1^{β/β}* mutant exhibited consistent midline crossing phenotype described previously (*Serafini et al., 1996*). For tdTomato labeling of DRG axons, pregnant dams at E9.5 were administered via oral gavage a dose of tamoxifen solution (0.5 mg / 50 µl) diluted in peanut oil that was warmed at 37 °C. Following dissection, tdTomato fluorescent expression was screened using an epifluorescent microscope.

## Immunohistochemistry, tissue clearing, and imaging

Embryos (E10.5-E11.5) were fixed overnight in 4% paraformaldehyde (PFA) in phosphate buffered saline (PBS). For wholemount immunostaining, embryo samples were pretreated with methanol dehydration, bleached in 10% hydrogen peroxide o/n at 4 °C, then subject to the reverse MeOH gradient re-rehydration treatment followed by staining based on either iDISCO or CUBIC protocols (*Renier et al., 2014*; *Susaki et al., 2014*; *Tainaka et al., 2014*).

For iDISCO based clearing and staining, re-hydrated samples were washed in 5% DMSO/0.3 M Glycine/PTxwH (1 X PBS, 0.01% Triton X-100, 0.05% Tween-20, 0.02% NaN3) sequentially for 1 and 2 hr; then washed in PTxwH x3 for 30 min each; and blocked in 3% Donkey Serum (DS)/PTxwH with shaking at 37 °C, o/n. Samples were then incubated with primary antibody diluted in blocking solution for 3 days at 37 °C; washes done in PTxwH for 1 hr x2, 2 hr x2, then o/n and 1 day; followed by incubation of secondary antibody diluted in blocking solution o/n at 37 °C. Washes were done the next day in PTxwH for 1 hr x2, 2 hr x2, then o/n and 1 day. Next, embryos were washed in PBS shaking at RT, 1 hr x2, 2 hr x2, then o/n. For clearing, samples were dehydrated in the MeOH gradient, then washed 3 x in 100% MeOH for 15 min each, transferred to 100% dicholoromethane (DCM, Sigma-Aldrich 270997–1 L), and finally incubated in DiBenzyl Ether (DBE, Sigma-Aldrich 108014–1 KG) until clear (*Renier et al., 2014*).

For CUBIC based clearing and staining, embryos were washed after rehydration in 0.1% Triton X-100 for 10–15 min x3, and then transferred to blocking buffer (5% heat-inactivated normal goat serum, 20% DMSO in PBS), at 37 °C o/n. Samples were incubated with primary antibodies diluted in blocking buffer for 3 days at 37 °C, then washed in 20%DMSO in PBS for 1 hr x6. Secondary antibodies in blocking solution were incubated o/n at 37 °C. Embryos were washed 1 hr x6 and then transferred to CUBIC1 solution (25 wt% N,N,N′,N′-tetrakis(2-hydroxypropyl)ethylenediamine (Sigma 122262), 25 wt% urea and 15 wt% Triton X-100) for clearing. Following successful clearing, samples were equilibrated in CUBIC2 solution (50 wt% sucrose, 25 wt% urea, 10 wt% triethanolamine (Sigma T58300), and 0.1% (v/v) Triton X-100) for imaging (*Susaki et al., 2014*; *Tainaka et al., 2014*).

Imaging of cleared embryos was done on a Leica SP8 laser scanning confocal microscope using a 10 X objective (NA = 1.05). Inverted fluorescent images are 2D projections of confocal sections of the entire stack or the sections surrounding the region of interest.

For immunostaining on tissue sections, PFA-fixed embryos were equilibrated with 30% sucrose in PBS and frozen in the Tissue-Tek OCT embedding medium. Thin sections (16 µm) were cut on a cryostat (Leica) and processed for antibody staining (*Ma and Tessier-Lavigne, 2007*). Cryosections were permeabilized and blocked in blocking buffer (0.1% Triton X-100 and 10% goat serum in PBS) for 1 hr at room temperature and overlaid with primary antibodies diluted in blocking buffer overnight at 4 °C. Sections were then washed with PBS and 0.1% Triton X-100 x3, blocked again for 2 hr at room temperature and then incubated with secondary antibodies diluted in blocking buffer overnight at 4 °C. Sections were then washed with PBS x3 and sealed with cover glass before imaging.

Fluorescently labeled tissue sections were imaged on a wide field fluorescence microscope (Zeiss, Inc) or laser confocal systems (Leica SP8 system).

To detect tdTomato expression, rabbit anti Red Fluorescent Protein (RFP) polyclonal antibodies (1:500) were used. For NF staining, a monoclonal mouse antibody (2H3, 1:200) was used. Cy3- or Cy5-conjugated secondary antibodies (1:500) were used.

HRP staining in wholemount embryos followed previously described method (*Ma and Tessier-Lavigne, 2007*). Embryos were bleached with hydrogen peroxide overnight, incubated with primary antibodies and then HRP-conjugated secondary antibodies. After converting the 3,3'-diaminobenzidine (DAB) substrate to brownish deposit around labeled axons, the embryos were cleared in a BA/BB solution (benzylalcohol:benzylbenzonate, 2:1) (*Huber et al., 2005*), and imaged by the Spot II-RT camera mounted on a Stemi-6 stereoscope (Zeiss, Inc).

## DiI labeling of DRG axons in the spinal cord

For single neuron labeling by DiI, embryos were fixed in 4% PFA and then cut open from the ventral side to expose the DRG. A small dye crystal was delivered to the DRG from an ethanol solution of DiI (0.5%) in a glass pipette (1 μm opening) by iontophoresis with a current of ~5–20 mA as previously described (*Ma and Tessier-Lavigne, 2007*). The dye was allowed to diffuse at 25 °C overnight. The floorplate of the spinal cord was cut to create an open-book, which was then laid down in an imaging chamber with the lateral side down. The labeled axons were imaged on a Leica SP8 confocal microscope at 20 X objective (NA = .4). Maximum projections were traced and skeletonized in ImageJ using the Neurotrace plugin.

## In situ hybridization

Cryosections (16 μm) of E10.5 and E11.5 mouse embryos were processed for in situ analysis following a published procedure (*Ma and Tessier-Lavigne, 2007*). [35]S-labeled probes were generated using the following published templates for *Netrin-1* (*Serafini et al., 1996*). Dark field images were taken on a Zeiss compound microscope.

## Experimental design and statistical analysis

Embryos with different genotypes were collected from the same litters and subjected to the immunohistochemical analysis described above. Image comparison was done in the same spinal cord region of embryos with the same age and similar body size. The number of animals analyzed is listed in the figure legends.

To quantify misprojections in wholemount embryos, line scans of pixel intensity were taken from lateral side of wholemount embryos in ImageJ at three regions: proximal (right above the DREZ), medial (75 μm above the DREZ), and distal (150 μm above the DREZ and closest to the dorsal midline). After background subtraction, the peaks representing the labeled axons were counted above each brachial DRG at different regions and used for comparison by one-way ANOVA.

Quantification of misprojections in cross sections was done with the NeuronJ plugin in ImageJ. To quantify horizontal misprojections, the total length of axons projecting centrally were traced and measured on each hemisection. To quantify dorsal misprojections, the distance from the midline to the tip of dorsally projecting axons was measured following the curve of the cross-section. Values for multiple hemisections per embryo were summed and then averaged for statistical comparison using one-way ANOVA with post-hoc Tukey's test. For each animal, lengths of horizontal misprojections and the distance between DREZ to midline were averaged from two to four sections.

To quantify DiI labeled single axons, clearly visible axons were first divided into two groups, single or bifurcated. For single axons, we used the neighboring axons as a reference to determine whether they turned into the DREZ. Bifurcated axons that grow in the DREZ are considered normal and those projecting away from the DREZ were considered misguided. Normal and misguided axons were tallied from each spinal cord and used for comparison by t-tests. For quantification of the branching angle for DiI analysis a circle with the radius of 75 μm from the branch junction was determined and the angle formed from the main axon shaft and the extending branch was measured. An ANOVA between the varying genotypes was done with Tukey post-hoc test for determining significance.

For all statistical analysis, a minimum of three mice were analyzed. All data are expressed as Mean ± Standard Error of the Mean (SEM), and statistical values are included in figure legends.

## Note

Andrea Yung is now affiliated with Genentech; all her work for this manuscript was conducted while affiliated with Harvard Medical School.

## Acknowledgements

We thank Zongxiu Zhang and Yonghong Zhou for mouse management. We also thank members of the Ma lab for helpful discussion throughout the study, and Drs. Stephen Tymanskyj and Matthew Dalva for comments on the early version of the manuscript. This work was supported by grants from the National Institutes of Health to KN (F31NS108671), LVG (R21DC014916), AJ (R01NS095908), and LM (R01NS062047 and R01NS112504).

## Additional information

### Funding

| Funder | Grant reference number | Author |
|---|---|---|
| National Institute of Neurological Disorders and Stroke | F31NS108671 | Kelsey R Nickerson |
| National Institute on Deafness and Other Communication Disorders | R21DC014916 | Lisa V Goodrich |
| National Institute of Neurological Disorders and Stroke | R01NS095908 | Alexander Jaworski |
| National Institute of Neurological Disorders and Stroke | R01NS062047 | Le Ma |
| National Institute of Neurological Disorders and Stroke | R01NS112504 | Le Ma |

The funders had no role in study design, data collection and interpretation, or the decision to submit the work for publication.

### Author contributions

Bridget M Curran, Data curation, Formal analysis, Investigation, Visualization, Methodology, Writing - original draft, Project administration, Writing - review and editing; Kelsey R Nickerson, Data curation, Formal analysis, Funding acquisition, Methodology; Andrea R Yung, Resources; Lisa V Goodrich, Resources, Funding acquisition, Writing - review and editing; Alexander Jaworski, Supervision, Funding acquisition, Writing - review and editing; Marc Tessier-Lavigne, Resources, Writing - review and editing; Le Ma, Conceptualization, Resources, Formal analysis, Supervision, Funding acquisition, Investigation, Methodology, Writing - original draft, Project administration, Writing - review and editing

### Author ORCIDs

Lisa V Goodrich ⓘ https://orcid.org/0000-0002-3331-8600
Alexander Jaworski ⓘ https://orcid.org/0000-0002-4596-5450
Le Ma ⓘ https://orcid.org/0000-0003-2769-9416

### Ethics

All animal procedures followed the Guidelines for the Care and Use of Laboratory Animals of the National Institutes of Health and the approved IACUC protocols of the Thomas Jefferson University (#01558 and #01559) and of the Brown University (#21-12-0006).

Reviewer #1 (Public review): https://doi.org/10.7554/eLife.94109.3.sa1

Reviewer #2 (Public review): https://doi.org/10.7554/eLife.94109.3.sa2
Reviewer #3 (Public review): https://doi.org/10.7554/eLife.94109.3.sa3
Author response https://doi.org/10.7554/eLife.94109.3.sa4

## Additional files

### Supplementary files
• MDAR checklist

### Data availability
All numerical data generated and analyzed are included in the manuscript as source data file attached to each figure.

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
