## [Editor Report · eLife assessment]

This **important** study expands our understanding of the role of two axon guidance factors in a specific axon guidance decision. The strength of the study is the **compelling** axonal labeling and quantification, which allows the authors to establish precise consequences of the loss of each guidance factor or receptor.

---

## [Referee Report · Reviewer #1 (Public review)]

Summary:

The current manuscript provides an extensive in vivo analysis of two guidance pathways identifying multiple mechanisms that shape the bifurcation of DRG axons when forming the dorsal funiculus in the DREZ.

Strengths:

Multiple mouse mutant lines were used, together with complementary techniques; the results are very clear and compelling.

The findings are very significant and clearly move forward our understanding of the regulation of axonal development at the DREZ.

Weaknesses:

No major weaknesses were found. As it is I have no recommendations that would increase the clarity or quality of the manuscript.

---

## [Referee Report · Reviewer #2 (Public review)]

Summary:

In this manuscript, the authors conduct a detailed analysis of the molecular cues that control guidance of bifurcated dorsal root ganglion axons in a key region of the spinal cord called the dorsal funiculus. This is a specific case of axon guidance that occurs in a precise way. The authors knew that Slit was important but many axons still target correctly in Slit knockouts, suggesting a role for other guidance factors. Netrin1 is also expressed in this region, so they looked at netrin mutants. The authors found axons outside the DREZ in the Ntn1 mutants, and they show by single neuron genetic labeling that many of these come from DRG neurons. Quantified axonal tracing studies in Slit1/2, Ntn1, or triple mutant embryos supports the idea that Slit and Ntr1 have distinct functions in guidance and that the effect of their loss is additive. Interestingly none of these knockouts affect bifurcation itself but rather the guidance of one or both of the bifurcated axon terminals. Knockout of the Slit receptors (Robo1/2) or the Netrin 1 receptor (DCC) in embryos causes similar guidance defects to loss of the ligands, providing an additional confirmation of the requirement for both guidance pathways. This study expands understanding of the role of the axon guidance factors Ntr1/DCC and Slit/Robo in a specific axon guidance decision. The strength of the study is the careful axonal labeling and quantification, which allows the authors to establish precise consequences of the loss of each guidance factor or receptor.

---

## [Referee Report · Reviewer #3 (Public review)]

Summary:

In this paper, Curran et al investigate the role of Ntn, Slit1 and Slit 2 in axon patterning of DRG neurons. The paper uses mouse genetics to perturb each guidance molecule and its corresponding receptor. Cre-based approaches and immunostaining of DRG neurons are used to assess the phenotypes. Overall, the study uses the strength of mouse genetics and imaging to reveal new genetic modifiers of DRG axons. The conclusions of the experiments match the presented results. The paper is an important contribution to the field, as evidence that dorsal funiculus formation is impacted by Ntn and Slit signaling. The paper clearly demonstrates molecules that impact the patterning of the dorsal funiculus formation, which can provide a foundation for future studies on the specific steps in that patterning that require the studied molecules.

Strengths:

The manuscript uses the advantage of mouse genetics to investigate axon patterning of DRG neurons. The work does a great job of assessing individual phenotypes in single and double mutants. This reveals an intriguing cooperative and independent function of Ntn, Slit1 and Slit2 in DRG axon patterning. The sophisticated triple mutant analysis is lauded and provides important insight.

Weaknesses:

Overall, the manuscript is sound in technique and analysis. While not a weakness, the paper provides the foundation for future studies that investigate the specific molecular mechanisms of each step in the patterning of the dorsal funiculus.

---

## [Author Response]

The following is the authors’ response to the original reviews.

**Public Reviews:**

**Reviewer #1 (Public Review):**
Summary:The current manuscript provides an extensive in vivo analysis of two guidance pathways identifying multiple mechanisms that shape the bifurcation of DRG axons when forming the dorsal funiculus in the DREZ.Strengths:Multiple mouse mutant lines were used, together with complementary techniques; the results are very clear and compelling.The findings are very significant and clearly move forward our understanding of the regulation of axonal development at the DREZ.Weaknesses:No major weaknesses were found. As it is I have no recommendations that would increase the clarity or quality of the manuscript.
**Reviewer #2 (Public Review):**
Summary:In this manuscript, the authors conduct a detailed analysis of the molecular cues that control the guidance of bifurcated dorsal root ganglion axons in a key region of the spinal cord called the dorsal funiculus. This is a specific case of axon guidance that occurs in a precise way. The authors knew that Slit was important but many axons still target correctly in Slit knockouts, suggesting a role for other guidance factors. Netrin1 is also expressed in this region, so they looked at netrin mutants. The authors found axons outside the DREZ in the Ntn1 mutants, and they show by single-neuron genetic labeling that many of these come from DRG neurons. Quantified axonal tracing studies in Slit1/2, Ntn1, or triple mutant embryos support the idea that Slit and Ntr1 have distinct functions in guidance and that the effect of their loss is additive. Interestingly none of these knockouts affect bifurcation itself but rather the guidance of one or both of the bifurcated axon terminals. Knockout of the Slit receptors (Robo1/2) or the Netrin 1 receptor (DCC) in embryos causes similar guidance defects to loss of the ligands, providing additional confirmation of the requirement for both guidance pathways.Strengths:This study expands understanding of the role of the axon guidance factors Ntr1/DCC and Slit/Robo in a specific axon guidance decision. The strength of the study is the careful axonal labeling and quantification, which allows the authors to establish precise consequences of the loss of each guidance factor or receptor.Weaknesses:There are some places in the text where the discussion of these data is compared with other studies and models, but additional details would help clarify the arguments.

The details were added to the first section of Discussion in the revision to address this weakness. Also see the response to the recommendations below.

**Reviewer #3 (Public Review):**
Summary:In this paper, Curran et al investigate the role of Ntn, Slit1, and Slit 2 in the axon patterning of DRG neurons. The paper uses mouse genetics to perturb each guidance molecule and its corresponding receptor. Cre-based approaches and immunostaining of DRG neurons are used to assess the phenotypes. Overall, the study uses the strength of mouse genetics and imaging to reveal new genetic modifiers of DRG axons. The conclusions of the experiments match the presented results. The paper is an important contribution to the field, as evidence that dorsal funiculus formation is impacted by Ntn and Slit signaling. However, there are some potential areas of the manuscript that should be edited to better match the results with the conclusions of the work.Strengths:The manuscript uses the advantage of mouse genetics to investigate the axon patterning of DRG neurons. The work does a great job of assessing individual phenotypes in single and double mutants. This reveals an intriguing cooperative and independent function of Ntn, Slit1, and Slit2 in DRG axon patterning. The sophisticated triple mutant analysis is lauded and provides important insight.Weaknesses:Overall, the manuscript is sound in technique and analysis. However, the majority of the manuscript is about the dorsal funiculus and not the bifurcation of the axons, as the title would make a reader believe. Further, the manuscript would provide a more scholarly discussion of the current knowledge of DRG axon patterning and how their work fits into that knowledge.

We revised the title as suggested. Additional discussion of DRG axon growth at the DREZ is added to the last section of the Discussion in the revision. Also see the response to the recommendations below.

**Recommendations for the authors:**

**Reviewer #1 (Recommendations For The Authors):**
Given the reasons stated above, I have no specific recommendations for the authors.There is a typo in the Abstract (... mice with triple deletion of Ntn1, Slit2, and Slit2....).

Corrected in the revision.

**Reviewer #2 (Recommendations For The Authors):**
(1) The authors twice repeated that their data on DRG guidance defects in the Ntn1 mutants differ from studies previously published in references 19 and 26. However it is unclear to me, without having read those other studies, what is actually different between this study and those, and why there would be differences between the results from two groups. If the authors think this is an important point to make they need to more clearly say what the other group saw and offer an explanation of why the data may be different.

We added detailed comparison of the defects from different studies to the first section of the Discussion and suggested multiple roles of Ntn1 in controlling sensory axon growth at the DREZ in the revision.

(2) In the final section of the discussion it says, "The guidance regulation of DRG axon bifurcation by Slit and Ntn1 may be similar to but overshadowed by their function in midline guidance [43]." The meaning of this sentence was unclear to me. I had been thinking that since there are total knockout embryos (not conditional) there could be patterning effects that happen before the DRG branching that influence the formation of the DREZ. Is this what the authors mean to say here? How can the authors show that the guidance factors they have knocked out are actually functioning in the DRG neurons?

We agree with the reviewer that the first sentence is vague, so we edited the paragraph and included the discussion of the regulation of DRG axons at the DREZ, which was the main theme of this last section. In addition, we agree with the reviewer’s suggestion of the possible indirect role of Ntn1 on DRG axons via the control of interneuron migration. This possibility was included in the last paragraph of the Discussion.

(3) In several of the figures (3T, 5I, 5J) there are distance measurements that are presumably averages of multiple axons in 3 or 4 embryos because 3-4 points are shown per graph. However, the figure and methods do not say how many axons were measured per embryo and I could not find if it says these numbers are averages. Clarifying the details of these panels would be useful.

The n is the number of animals analyzed and is now added to the figure legends. From each animal, multiple sections (2-4) were analyzed for various parameters in Fig. 3 and 5. This information was added to the Method section of the revision.

**Reviewer #3 (Recommendations For The Authors):**
Overall the data matches the conclusions in the paper. However, to this reviewer, the title suggests that Ntn and Slit will have defects in bifurcation. This is not the presented phenotype. I recommend the authors change the title to better reflect the findings of the work.

We edited the title of the revised manuscript to reflect the control of growth direction in the context of bifurcation.

The introduction of the work clearly outlines what is known about DREZ formation in mice but could extend its discussion to other systems like chick and zebrafish (Jaeda Coutinho-Budd et al. 2008, Wang and Scott 2000, Golding et al 1997, Nichols and Smith 2019, Kikel-Coury et al 2021). These studies are particularly important given that pioneer events, including bifurcation, can be visualized. Acknowledging the contribution of other model systems to the understanding of DRG axon patterning is important to improve the scholarly discussion of the paper.

We added more detailed discussion of the current knowledge of DRG axon growth at the DREZ from several relevant studies of the rodent and zebrafish models in the last section of Discussion.

In the data presented, the authors see defects in the axon patterning of DRG neurons and conclude it is a defect in the dorsal funiculus formation. Another interpretation is that a subset of axons cannot invade the spinal cord boundary properly. This phenotype was observed in zebrafish with timelapse imaging (Kikel-Coury et al 2021). It may not be necessary to specifically test the axons' ability to enter the spinal cord in this paper, but the possibility that this could drive the presented phenotypes should be more clearly stated in the results. Entry is not thoroughly addressed in this paper and would need to be confirmed by labeling the edge of the spinal cord with a second reporter. No entry would obviously impact axon targeting. However, delayed entry could place the axon in a navigation environment that is atypical, causing it to navigate aberrantly and present as a funiculus phenotype.

We thank the reviewer for raising this very interesting point. In our present view, dorsal funiculus formation is related to DRG axon patterning, which involves growth, guidance, and bifurcation of the incoming afferents at the dorsal spinal cord. We believe that these events are highly coordinated by various environmental cues to generate the DREZ and the dorsal funiculus. The defects we observed could result from the disruption of such coordination that leads to misregulation of DRG axon entry at the dorsal spinal cord, as suggested by the reviewer. We propose that further analysis by time-lapse imaging as done in zebrafish would provide better understanding of such coordination. This discussion was included in the last section of Discussion.

The authors should clarify that their approach does not knock out molecules in a cell-specific way. This would specifically impact the interpretation of the Dcc phenotypes. It is possible that UNC-40/DCC is guiding cells that are not labeled. The non-autonomous role of UNC-40/DCC should be clearly stated as a possibility.

This discussion was added to the last paragraph of the Discussion section.